# Therapeutic Targeting of Cancer Stem Cells in Human Glioblastoma by Manipulating the Renin-Angiotensin System

**DOI:** 10.3390/cells8111364

**Published:** 2019-10-31

**Authors:** David C. H. Tan, Imogen M. Roth, Agadha C. Wickremesekera, Paul F. Davis, Andrew H. Kaye, Theo Mantamadiotis, Stanley S. Stylli, Swee T. Tan

**Affiliations:** 1Department of Neurosurgery, Wellington Regional Hospital, Wellington 6021, New Zealand; Tan.David@ccdhb.org.nz (D.C.H.T.); Agadha.Wickremesekera@ccdhb.org.nz (A.C.W.); 2Gillies McIndoe Research Institute, Wellington 6021, New Zealand; imogen.roth@gmri.org.nz (I.M.R.); paul.davis@gmri.org.nz (P.F.D.); 3Department of Surgery, The University of Melbourne, Parkville, VIC 3050, Australia; andrewk@hadassah.org.il (A.H.K.); Stanley.Stylli@mh.org.au (S.S.S.); 4Department of Neurosurgery, Hadassah Hebrew University Medical Centre, Jerusalem 91120, Israel; 5Department of Neurosurgery, The Royal Melbourne Hospital, Parkville, VIC 3050, Australia; 6Wellington Regional Plastic, Maxillofacial & Burns Unit, Hutt Hospital, Lower Hutt 5040, New Zealand

**Keywords:** glioblastoma, renin-angiotensin system, cancer stem cells, drug repurposing

## Abstract

Patients with glioblastoma (GB), a highly aggressive brain tumor, have a median survival of 14.6 months following neurosurgical resection and adjuvant chemoradiotherapy. Quiescent GB cancer stem cells (CSCs) invariably cause local recurrence. These GB CSCs can be identified by embryonic stem cell markers, express components of the renin-angiotensin system (RAS) and are associated with circulating CSCs. Despite the presence of circulating CSCs, GB patients rarely develop distant metastasis outside the central nervous system. This paper reviews the current literature on GB growth inhibition in relation to CSCs, circulating CSCs, the RAS and the novel therapeutic approach by repurposing drugs that target the RAS to improve overall symptom-free survival and maintain quality of life.

## 1. Introduction

Human astrocytic tumors are the most common primary intra-axial brain tumors. Under the World Health Organization (WHO) classification of central nervous system tumors, grade I astrocytomas include the more well-circumscribed pilocytic astrocytomas, in contrast to grade II to IV diffuse astrocytomas [1]. The presence of cytological atypia confers a grade II tumor. Anaplasia and mitotic activity confer a grade III tumor. Glioblastoma (GB), the most aggressive astrocytic tumor, classified as a grade IV astrocytoma, is characterized by microvascular proliferation and palisading necrosis. Treatment of GB traditionally involves maximal safe surgical resection for cytoreduction followed by adjuvant chemoradiotherapy with concomitant use of radiotherapy and the alkylating agent temozolomide, extending median survival to 14.6 months [2]. Methylation of the *O*^6^-methylguanine-DNA methyltransferase (MGMT) promoter is associated with better response to temozolomide and prolonged survival. Furthermore, the longstanding obstacle of the delivery of chemotherapy agents to the central nervous system due to the presence of the blood brain barrier may be overcome by a promising novel drug delivery system that was developed, involving curcumin-loaded chitosan polylactic-co-glycolic acid nanoparticles modified with sialic acid, to penetrate the blood brain barrier with anti-aldehyde dehydrogenase to target the CSCs [3].

The recent revision of the WHO classification of central nervous system tumors incorporates molecular parameters: a paradigm shift that provides dynamic phenotype and genotype classifications that impacts on prognosis and outcomes. Known intrinsic factors affecting the prognosis of GB include isocitrate dehydrogenase (IDH) mutation and methylation of the MGMT gene. GBs are divided into IDH-wildtype (90% of cases) and IDH-mutant tumors [1]. IDH is an enzyme involved in catalyzing oxidative decarboxylation of isocitrate to 2-oxoglutarate. The most common mutation in GB affects IDH1 with a single amino acid missense mutation at arginine 132 replaced by histidine (IDH1 R132H) [4]. IDH-wildtype GB tends to arise de novo, while IDH-mutants tend to progress from lower-grade precursor lesions and are commonly found in younger patients [5]. IDH mutants with methylation fingerprints [6] are associated with a better survival rate due to the accumulation of 2-hydroxyglutarate, secondary to loss of normal enzymatic function [7], increasing the sensitivity of the tumors to selective chemoradiotherapy [8]. Genetic alterations typical of IDH-wildtype GB include TERT promoter mutations (80%), loss of chromosome 10q (70%), homozygous deletion of CDKN2A/DKN2B (60%), loss of chromosome 10p (50%), EGFR alterations (55%), PTEN mutations (40%), TP53 mutations (25–30%), and PI3K mutations (25%) [1].

The original four GB subtype classification (proneural, neural, classical and mesenchymal) based on the genomic analysis of PDGFRA, IDH1, EGFR and NF1 coupled with a transcriptional profile by the Cancer Genome Atlas Network in 2010 [9], was recently refined to include three GB subtypes, namely classical, mesenchymal and proneural/neural [10,11]. Genomic and transcriptomic analysis demonstrate biological heterogeneity between different GB subtypes with important implications for future research. The poor survival rates of GB, together with the recent discovery of key molecular pathways regulating GB cell biology, fueled intense research to find novel therapeutic targets, particularly at the genomic and molecular levels.

## 2. Glioblastoma Cancer Stem Cells

Cancer stem cells (CSCs) in human brain tumors were initially discovered by the identification of cells expressing the cell surface marker CD133, a cell surface pentaspan transmembrane glycoprotein located in plasma membrane protrusions [12]. This observation was further extended by a study demonstrating stem-like neural precursor cells in GB, which can initiate growth and recurrence of the tumor even following multiple serial transplantations [13]. CSCs divide asymmetrically giving rise to identical, highly tumorigenic CSCs, and non-tumorigenic cancer cells which form the bulk of the tumor, contributing to intra-tumoral heterogeneity. The aggressive nature of GB is attributed to the presence of small subpopulations of CSCs and the potential molecular treatment options for targeting these GB CSCs were reviewed extensively [14]. Quiescent GB CSCs have the capacity for perpetual self-renewal and proliferation supported by tumor microenvironmental factors including TGF-β and hypoxia to promote tumor recurrence, providing a potential explanation for resistance to conventional treatments [15]. This ability for self-renewal is maintained by the Notch, Sonic hedgehog, and Wnt signaling pathways [16]. On the other hand, non-stem cancer cells can convert to CSCs due to epigenetic alterations conferring phenotypic plasticity to the glioma cell population. Recent evidence suggests that dynamic plasticity and bidirectional interconversion are possible in heterogenous tumor populations [17].

The CSC markers expressed in GB are categorized according to the cellular localization which include cell surface markers (e.g., CD133, CD15, A2B5, L1CAM), cytoskeletal proteins (e.g., nestin), transcription factors (e.g., SOX2, NANOG, OCT4), post-transcriptional factors (e.g., Musashi1), and polycomb transcriptional suppressors (e.g., Bmi1, Ezh2) [14]. Yamanaka et al. achieved a significant breakthrough with the discovery that mature mouse embryonic cells and adult fibroblasts can be reprogrammed to form pluripotent stem cells by adding a combination of key transcription factors OCT4, SOX2, c-MYC and KLF4 [18]. These factors are known to be expressed by embryonic stem cells (ESCs), and over-expression of these transcription factors can result in the transformation of somatic cells into induced pluripotent stem cells (iPSCs) [19,20]. Primitive populations expressing ESC markers such as NANOG4, KLF4, c-MYC, OCT4 and SOX2 were identified in GB [21]. Importantly, NANOG was identified as an independent prognostic factor in predicting survival for GB [22]. We previously proposed the presence of a CSC hierarchy in GB, implicating that OCT4+ cells represent the most primitive CSCs, which can differentiate to form SOX2+ and SALL4+ progenitor cells [21]. Invariant stem cell hierarchy is seen in GB with slow-cycling stem cells giving rise to fast-cycling progenitor cells which in turn generate non-proliferative cells, with the presence of outlier stem cells where chemotherapy facilitates proliferation of drug resistant stem cells [23,24].

Transcription factors including OCT4 and SOX2 may play a critical role in perpetual self-renewal of GB CSCs [14]. SOX2 which is highly expressed in GB [21] is considered a master transcription factor crucial in maintaining pluripotency of mammalian ESCs and is exponentially correlated with the expression of CD133 [25], a cell surface marker commonly seen in brain tumors as described above. SOX2 is shown to be crucial in maintaining plasticity for bidirectional conversion between cancer stem-like and differentiated glioma cells in patient-derived mouse xenografts [26]. In addition, SOX2 silencing in GB tumor-initiating cells was shown to inhibit tumor proliferation [27], providing a potential treatment strategy for GB at the cellular level [28]. For instance, tunicamycin, an inhibitor of N-linked glycosylation which acts as an endoplasmic reticulum stress inducer, was shown to cause cell cycle arrest in G1 phase, blocking the self-renewal capability of glioma CSCs by reducing the expression of SOX2 [29].

Traditionally, the contrast-enhancing components of GB seen on MRI were thought to be the moving front of tumor progression and invasion and as such were targeted for neurosurgical resection. However, multimodal MRI techniques such as diffusion tensor imaging coupled with magnetic resonance spectroscopy confirmed the presence of tumor cells beyond the contrast-enhancing rim [30]. These infiltrating tumor edges that show contrast enhancement harbor significantly higher percentages of CD133+ cells and are associated with a higher proliferative index [31]. Furthermore, tumor cells found in the normal brain beyond the margin of contrast enhancement, also show the presence of CD133+ and SOX2+ cells [32], confirming the infiltrative nature of GB and that these CSCs are a reservoir for the initiation of tumor recurrence following surgical resection and adjuvant chemoradiation.

The JAK-STAT3 signaling pathway is implicated in promoting self-renewal of GB CSCs. It involves the activation of JAK, phosphorylation of STAT proteins, and their translocation into the nucleus. STAT3 proteins are essential transcription factors in this signaling pathway. Pharmacological inhibition of the STAT3 activator JAK leads to decreased STAT3 transcriptional activation and reduced levels of associated matrix metalloproteinases (MMPs), potentially impacting on the extracellular matrix degrading ability of invadopodia [33], impeding the migratory and invasive potential of GB [34]. STAT3 binds to the Notch1 promoter leading to the activation of Notch signaling which also activates the transcription of stem cell markers in astrocytomas [35]. Inhibition of the Notch signaling pathway also impedes the maintenance of glioma stem cells and tumorsphere formation, in addition to reducing the expression of the glioma stem cell markers CD133, SOX2 and nestin [36]. From a therapeutic point of view, curcumin, a naturally occurring component of turmeric, was shown to inhibit JAK signaling, inducing reactive oxygen species, and down-regulating STAT3 phosphorylation, resulting in reduced proliferation of the tumor cells [37]. Curcumin-induced reactive oxygen species promote cytotoxicity, DNA damage and apoptosis [38]. Rather than relying only on the development of novel compounds, repurposing existing FDA-approved drugs to target GB would be a faster route to target oncogenic GB cell functions, as shown by targeting invadopodia activity in GB cell lines [39]. 

## 3. Circulating Cancer Stem Cells and Epithelial-to-Mesenchymal Transition

The concept of circulating CSCs and “liquid biopsy” was proposed as an alternative to obtaining histological specimens for diagnosis and molecular typing of the tumors [40]. It presents an alternative mechanism to explain the local recurrence of GB, implicating epithelial-to-mesenchymal (EMT) and mesenchymal-to-epithelial (MET) transformational pathways [41]. This paradigm is counterintuitive to the concept of activation of regional non-circulating quiescent GB CSCs in causing local recurrence of GB. Despite the invasive nature of GB and the presence of circulating CSCs, the reasons for the reported rarity of distant metastatic GB [42,43,44,45] remain unknown.

Historically, the concept of circulating CSCs is supported by studies demonstrating immunosuppressed patients who had received transplanted organs from donors with GB [46] and subsequently developed metastatic GB in lymph nodes and distant organs [47], and identification of circulating CSCs in peripheral blood of GB patients [48]. Early commentary on ultrastructural features suggested two potential factors that refute the possibility of circulating GB CSCs. Firstly, neoplastic glial cells are excluded from extravasation by the vascular basal laminae of the brain. Secondly, even if the neoplastic cells manage to escape into the vascular system, they are prevented from binding to the endothelium of the target organs, due to lack of appropriate cell adhesion molecules [49]. Another proposed reason is that the mesenchymal plasticity exhibited by GB CSCs is more differentiated and these CSCs are unable to find a suitable niche other than the brain [50].

More recently, EMT gained increased recognition and momentum as a process determining the presence or absence of metastases. Transcription factors and signaling pathways involved in EMT in gliomas were described [51]. Through EMT, an epithelial cell assumes increasing migratory ability and infiltrative capacity by transforming into a more immature mesenchymal cell type. The Hedgehog signaling pathway is shown to regulate the self-renewal of CD133+ glioma CSCs [52]. Activation of this pathway leads to increased expression of the transcription factors Snail and Slug, suppressing expression of E-cadherin, resulting in reduced junctional adherence between epithelial cells and increased capacity of cell migration [53]. GB cells were shown to be devoid of cell junctions while peri-tumoral cells display fully organized desmosomes and junctional complexes [54].

Therapeutically, nuciferine was shown to inhibit EMT by decreasing Slug expression via the AKT and STAT3 signaling pathways in GB [55]. In another study, a combination of an antagonist of the Hedgehog signal transducer Smoothened and an ATP competitor were shown to reduce the expression of Snail, Slug and Zeb1, thus inhibiting EMT, suggesting that combined inhibition of the PI3K/AKT/mTOR and Sonic Hedgehog pathways can be exploited together to suppress the growth of GB [56]. On the other hand, TGF-β1 was shown to induce EMT in GB cells by decreasing the expression of E-cadherin, inducing up-regulation of mesenchymal markers (e.g., N-cadherin, vimentin), crucial regulators (e.g., Twist1, β-catenin), EMT-activating transcription factors (e.g., Snail, Slug, Zeb1); and activating various downstream pathways including PI3K, Smads and MAP kinase [57]. An in vitro study showed that metformin inhibits TGF-β1 and suppresses the self-renewal capacity of GB CSCs and expression of CSC markers by decreasing the phosphorylation of AKT and mTOR [53]. Resveratrol, a natural phenol found in grapes, berries and peanuts, was also found to suppress EMT by suppressing the levels of MMPs and associated invadopodia activity, in addition to decreasing secondary gliosphere formation and expression of CSC markers via regulation of Smad-dependent signaling pathway [58].

The concept of circulating CSCs in GB introduces novel etiological pathways and may provide explanations for the resistance to traditional therapies and high rate of tumor recurrence. The regulatory effect of EMT by the renin-angiotensin system (RAS) was demonstrated in colorectal cancer. In one study, angiotensin II (ATII) was shown to induce migration of colorectal cancer cells via ATII receptor 1 (ATIIR1) and ATII receptor 2 (ATIIR2) [59]. Effects mediated by ATIIR1 are associated with changes typical of EMT, namely increased expression of E-cadherin, reduced ZEB1 and vimentin levels. A comprehensive review of the many current studies on GB CSCs and EMT-MET in glioma is beyond the scope of this review. However, further characterization may lead to the development of targeted systemic therapies based on the modulation of the RAS.

## 4. The Renin-Angiotensin System

The RAS (Figure 1) is a hormone system physiologically important in cardiovascular homeostasis and regulation of blood pressure in humans. Renin, which is physiologically secreted by the renal juxtaglomerular apparatus, acts to convert angiotensinogen, normally produced by the liver, to angiotensin I. Angiotensin I is then converted to ATII by angiotensin-converting enzyme (ACE), largely produced in the lungs. ATIIR1 and ATIIR2 are G-protein-coupled receptors with antagonistic effects. Activation of ATIIR1 induces cellular proliferation, inflammation and angiogenesis, whereas activation of ATIIR2 inhibits cell growth and enhances programmed cell death and cellular differentiation [60].

Renin is formed by the cleavage of its inactive precursor, pro-renin, to active renin, by binding to pro-renin receptor (PRR) [61], as well as by various enzymes including cathepsin B [62], cathepsin D and cathepsin G (Munro, M.; Wickremesekera, A.C.; et al. 2017) (Figure 1). COX-2 causes the up-regulation of PRR [63] (Figure 1). β-blockers reduce the production of pro-renin [64] (Figure 1). Insulin growth factor (IGF) activates insulin growth factor receptor-1 (IGFR-1) to promote conversion of pro-renin to active renin [65] (Figure 1). The action of ATII on ATIIR1 can be blocked by angiotensin receptor blockers (ARBs) [66] (Figure 1).

The RAS was implicated in the hallmarks of cancer [67,68]. We demonstrated the expression of components of the RAS: PRR, ACE, ATIIR1 and ATIIR2 by CSCs in different cancer types including head and neck cutaneous squamous cell carcinoma (SCC) [69], oral cavity SCC (OCSCC) affecting the lip [70], buccal mucosa [71] and oral tongue [72], liver metastases from colon adenocarcinoma [73] and metastatic melanoma to the brain [74]. More importantly, components of the RAS: PRR, ATIIR1 and ATIIR2 were shown to be expressed by the CSCs in GB; with ACE, PRR, ATIIR1 and ATIIR2 localizing to the endothelium of the microvessels [75] (Figure 2). These findings suggest that modulation of the RAS may provide novel therapeutic targeting of CSCs within GB and other types of cancers [76].

Lysosomal cysteine protease cathepsin B is increased six-fold in GB compared to normal brain tissues [77], which is further confirmed by studies demonstrating increased cathepsin B expression in GB, compared to anaplastic astrocytomas, low-grade gliomas and normal brain tissues [78,79]. Greater cathepsin B immunoreactivity in primary brain tumors and endothelial cells is associated with shorter survival time [80]. Another analysis reveals that cathepsin B and plasminogen activator inhibitor type 1 are important biomarkers for predicting overall survival of patients with GB [81]. Activation of cathepsins induces cell-membrane associated urokinase plasminogen activator (uPA), causing extracellular release of plasmin from plasminogen. Plasmin activates various MMPs capable of degrading basal lamina proteins [82], increasing the motility of glioma cancer cells. We undertook an analysis of GB-based studies within the online Oncomine^®^ platform for datasets that contained mRNA expression levels of cathepsin B. Oncomine [83] is an online tool that contains 715 mRNA and copy number expression datasets from 86,733 cancer and normal tissue samples (12,764 samples are normal tissue samples). Our datamining of the brain/central nervous system datasets deposited in the Oncomine Compendium examined the relative mRNA levels of cathepsin B in both GB and normal brain tissue. As shown by the data presented in Table 1, there is an elevation of cathepsin B in GB tissue, relative to normal brain in three studies (TCGA Brain, Bredel Brain 2, Sun Brain) [84].

Up-regulation of cathepsin B and uPA receptors induces SOX2 and Bmi1 expression, both critical for maintaining the stemness of glioma CSCs, while knockdown of cathepsin B and uPA receptors suppresses expression of SOX2, Bmi1 and nestin, in vivo [85]. Caffeine was found to suppress proliferation of GB cell lines, and is associated with decreased activity of cathepsin B and up-regulation of tissue inhibitor of metalloproteinase-1 via the MAPK signaling pathway [86]. RNA sequencing of a radio-resistant pediatric GB cell line following radiation revealed the over-expression of pro-cathepsin B, implicating the potential for alternative therapies that target metalloproteinases or cathepsin B [87]. Expression of cathepsin B and cathepsin D was demonstrated in OCT4+ and SALL4+ CSCs in IDH-wildtype GB [88] (Figure 2). These cathepsins constitute bypass loops of the RAS, contributing to the production of RAS peptides which promote proliferation of CSCs in GB. Therefore, targeting the RAS and its bypass loops in GB CSCs may potentially control the growth of GB tumors.

## 5. Repurposing Drugs that Target the RAS

Numerous drugs were demonstrated to promote GB cell apoptosis in vitro and in vivo by modulating the RAS [89,90]. ACE inhibitors reduce production of ATII, while ARBs selectively block ATIIR1 (Figure 1). The anti-neoplastic action of the RAS-modulating drugs is primarily due to the inhibition of ATII [91]. The ARB losartan, a selective inhibitor of ATIIR1, was shown to suppress growth of C6 rat glioma and induce apoptosis in C6 glioma cells [91]. Nonetheless, the ASTER study, a randomized placebo-controlled trial investigating the addition of losartan to the standard of care (concomitant use of radiotherapy and temozolomide) for patients with GB fails to show a difference in steroid requirement or significant improvement in median overall survival in patients with newly diagnosed GB [92]. Other studies show that selective synthetic renin inhibitors decrease DNA synthesis and induce apoptosis in GB cells [93], and that ARBs are associated with statistically improved progression-free survival and overall survival in 81 patients with GB [94].

Auranofin, an inhibitor of cathepsin B, and captopril, an ACE inhibitor, are included in the coordinated undermining of survival paths (CUSP9) treatment protocol—a trial targeting recurrent GB by combining nine repurposed drugs with temozolomide, highlighting the six themes important to cancer therapy, accepting that cytotoxic drugs alone are futile in prolonging survival of GB patients and these drugs may improve the efficacy of chemotherapeutic agents such as temozolomide [95]. The study combines the use of aprepitant, artesunate, auranofin, captopril, celecoxib, disulfiram, itraconazole, ritanvir, and sertraline, in conjunction with temozolomide. The initial results indicated little toxicity, maintenance of good quality of life, and hints of effectiveness. Experimental studies using patient-derived GB CSC cultures show increased sensitivity of CSCs to the drug combination with temozolomide, compared to temozolomide monotherapy [96].

Numerous epidemiological studies demonstrate a lower incidence of cancer and/or improved survival rate of cancer patients taking medications that modulate the RAS. These include a one-third reduction of the risk of developing skin SCC in patients who were treated with ACE inhibitors or ARBs [97] and a reduced risk of developing head and neck, gastric, colon and prostate cancers in patients receiving propranolol [98]. Treatment with aspirin, a COX-1 and COX-2 inhibitor [99] and ketorolac, a specific COX-2 inhibitor, [100], are associated with a reduction in the risk of developing bowel cancer [101] and reduction of recurrence and death in breast cancer patients [102] respectively. More recently, a reformulated “liquid” aspirin (IP1867B), an inhibitor of COX1/COX2 as well as IGF/IGFR-1 (signaling pathways that converge on the RAS) (Figure 1) was shown to reduce high-grade glioma tumor burden with an improved gastric side-effect profile [103]. Improved survival was observed in ovarian cancer patients who are administered non-selective β-blockers [104], and patients with multiple myeloma receiving propranolol [105]. Cathepsin B over-expression is associated with higher tumor grades and reduced overall survival in patients with OCSCC [106]. Importantly, improved survival of OCSCC patients after administration of curcumin, an inhibitor of cathepsin B, was reported [107]. More specifically, a recent study shows that the use of RAS inhibitors is associated with survival benefit in glioma patients [108].

Repurposing drugs, including anti-depressants, anti-convulsants, anti-hypertensives, statins, singly or in combination for the treatment of GB was recently reviewed [109] with positive effects. The understanding of the regulation of the RAS and CSCs in GB, in particular the expression and function of cathepsin B [110] and the IGF/IGFR-1 pathway [111], leads us to propose modulating the RAS, a singular systemic homeostatic pathway, using a combination of drugs (Figure 1), to simultaneously inhibit key steps of the RAS, its bypass loops and crosstalk signaling pathways interacting with the RAS. This may offer a novel therapeutic approach for patients with GB [76] to potentially increase overall survival while preserving their quality of life and avoiding drug toxicities. Currently we are undertaking a drug repurposing study using a cocktail consisting of propranolol (a β-blocker), metformin (an IGF/IGFR-1 blocker), curcumin (a cathepsin B blocker), aliskiren (a renin blocker), cilazapril (an ACE inhibitor), and losartan (an ARB) to treat GB [112].

A summary of current publications on therapeutic targeting of the RAS in GB, listed chronologically, is presented in Table 2.

## 6. Conclusions

The prognosis for patients with GB remains poor despite intensive research over the last 50 years. New therapeutic regimens are necessary to improve the overall survival and the quality of life of these patients. Further research into CSCs and the role of the RAS and its bypass loops and signaling pathways that converge onto the RAS, in the regulation of the CSCs in cancer, may underscore a potential paradigm shift in the treatment of GB. Randomized controlled trials incorporating repurposed drugs targeting these mechanisms are needed to demonstrate the efficacy of this novel therapeutic approach that may enhance the results of current treatment protocols.

## Figures and Tables

**Figure 1 cells-08-01364-f001:**
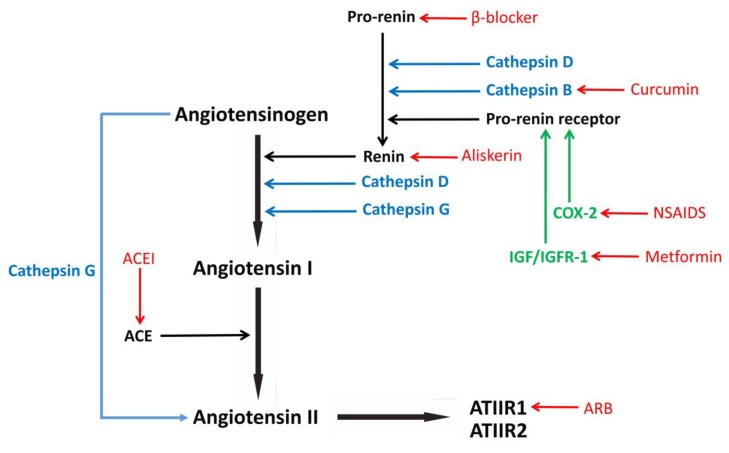
The renin-angiotensin system (RAS), its bypass loops and convergent signaling pathways, and medications that target key steps of these pathways. The classical RAS, highlighted in black, regulates blood pressure, stem cells and tumor development. Bypass loops of the RAS, highlighted in blue, involves enzymes such as cathepsins B, D and G provide redundancy, while other signaling pathways such as the COX-2 pathway and the IGF/IGFR-1 pathway, highlighted in green, converge on the RAS, to activate the pro-renin receptor. Key steps of the RAS and related pathways can be inhibited by commonly available medications, highlighted in red. Angiotensinogen (AGN) is physiologically synthesized and released by the liver and is cleaved by renin which is released by the kidneys, to form angiotensin I (ATI). Renin is formed following binding of pro-renin to the pro-renin receptor. Production pro-renin is reduced by β-blockers, and renin can be directly blocked using aliskerin. ATI is converted to angiotensin II (ATII) by angiotensin-converting enzyme (ACE), normally produced by the lungs. ACE can be blocked using ACE inhibitors (ACEI). ATII interacts with the G-protein coupled receptors ATII receptor 1 (ATIIR1) and ATII receptor 2 (ATIIR2), to restore homeostasis. ATIIR1 can be blocked using an ATIIR1 blocker (ARB). Cathepsins B and D are also renin-activating enzymes that convert pro-renin to renin. Curcumin inhibits the activities of cathepsin B. Cathepsin D also converts AGN to ATI, and cathepsin G converts ATI to ATII or AGN directly to ATII. The COX-2 pathway and the IGF/IGFR-1 pathway can be blocked using non-steroidal anti-inflammatory drugs (NSAIDS) and metformin, respectively.

**Figure 2 cells-08-01364-f002:**
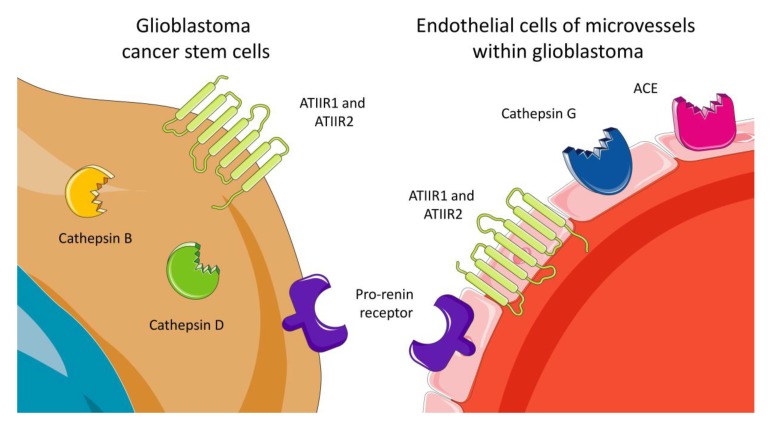
Expression of components of the renin-angiotensin system and proteins that constitute bypass loops of the renin-angiotensin system by cancer stem cells and the microvessels within glioblastoma. Cancer stem cells in glioblastoma express ATIIR1, ATIIR2, pro-renin receptor, cathepsin B and cathepsin D. The endothelium (pink cells) on the microvessels within glioblastoma express ACE, ATIIR1, ATIIR2 and cathepsin G.

**Table 1 cells-08-01364-t001:** Cathepsin B over-expression in glioblastoma compared to normal brain.

Number of Glioblastoma Samples	Number of Corresponding Normal Brain Samples	Total Number of Measured Genes	Mean Fold Change (Log2)	*p*-Value	Sample Type	Platform	Study
542	10	12,624	2.0662	1.96 × 10^-8^	mRNA	Human Genome U133A Array	TCGA Brain
27	4	14,836	1.819	1.84 × 10^-5^	mRNA	Not Defined	Bredel Brain 2
81	23	19,574	1.543	4.02 × 10^-7^	mRNA	Human Genome U133 Plus 2.0 Array	Sun Brain

Cathepsin B mRNA expression was examined in glioblastoma tissue within the Oncomine database. Displayed in the table are the mean fold changes vs. corresponding normal tissue in each study and overall *p*-value. Gene expression data are log transformed and normalized as previously described (Rhodes et al., 2004).

**Table 2 cells-08-01364-t002:** Summary of current publications on therapeutic targeting of the RAS in GB.

Authors	Year	Subjects	Medications	Effects
Rivera, et al.	2001	C6 rat glioma	Losartan	Reduction in tumor volume, decreased vascular density, mitotic index, cell proliferation
Juillerat-Jeanneret, et al.	2004	Human GB cell cultures	Renin inhibitors	Induced apoptosis in human glioblastoma cells
Arrieta, et al.	2005	C6 rat glioma	Losartan	Decreased tumor volume, induction of apoptosis in dose-dependent manner
Januel, et al.	2015	GB patients	ACEIs, ARBs	Improved progression-free survival and overall survival in multivariate analysis
Levin, et al.	2017	GB patients	Angiotensin system inhibitors (not specified) +/− bevacizumab	Improved survival, further survival advantage when renin-angiotensin system inhibitors were combined with low-dose bevacizumab
Mihajluk, et al.	2019	Human GB cell cultures	Reformulated aspirin (IP1867B)	Reduction in high-grade glioma cell viability, suppressed IL6/STAT3 and NF-κB networks, reduction in IGF1 and EGFR expression, less gastrointestinal side effects compared to conventional aspirin
Ramirez-Exposito, et al.	2019	Human neuroblastoma NB69, astroglioma U373-MG	Doxazosin	Concentration-dependent inhibition of cell growth, modification of proteolytic regulatory enzymes of RAS cascade
Skaga, et al.	2019	Human GB stem cell cultures	Aprepitants, auranofin, captopril, celecoxib, disulfiram, itraconazole, minocycline, quetiapine, sertraline	The combination effect of CUSP9 with temozolomide was superior to temozolomide monotherapy in clinical plasma concentrations
Ursu, et al.	2019	GB patients	Losartan	No difference in steroid requirement to reduce peritumoral edema

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
