# Peer review of "Therapeutic Targeting of Cancer Stem Cells in Human Glioblastoma by Manipulating the Renin-Angiotensin System"

_cells, 2019, doi:10.3390/cells8111364_

Round 1

Reviewer 1 Report

In this review article the authors do a thorough summary of the current literature regarding targeting the renin-angiotensin system (RAS) in order to kill high grade glioma (HGG) cancer stem cells (CSC). They particularly focus in repurposing drugs. Overall the manuscript is well structured and the theme is pertinent. However its novelty is average, since even the authors state: “the potential molecular treatment options for targeting these GB CSCs have been reviewed extensively (Kalkan 2015).” and as such they focus the review to a sub-set of HGG CSCs treatment. The manuscript is well written, however some sentences need to be re-worked for clarity. The authors are experts in the field, particularly studying the RAS in CSCs, and have published many research manuscripts related to this theme. In saying this they do acknowledge and discuss the research work performed by other groups that contributed to the advancement of this field. The authors state “Currently, we are undertaking a drug repurposing study using a cocktail consisting of propranolol (a β-blocker), metformin (an IGF/IGFR-1 blocker), curcumin (a cathepsin B blocker), aliskiren (a renin blocker) and cilazapril (an ACE inhibitor), losartan (an ATRB) to treat GB (Tan 2018).”, this raises the question: should this review be done after this research work has been published? The figures are very well structured, clear and of good quality. For the aforementioned reasons I think that the manuscript is appropriate for publication in the journal Cells after minor revisions. I would like to invite the authors to address the comments bellow:

Comments:

The authors use the old nomenclature, namely glioblastoma instead of the new nomenclature, high grade glioma, this should be changed throughout the manuscript.

Lines 35-37: Where it reads: “Glioblastoma (GB), the most aggressive astrocytic tumor, classified as a grade IV astrocytoma, is characterized by microvascular proliferation and palisading necrosis.” It should read: High grade glioma (HGG), the most aggressive astrocytic tumor, classified as a grade IV astrocytoma, is characterized by microvascular proliferation and a palisading invasive front.

Lines 155-157: “It presents an alternative mechanism local recurrence of GB, implicating epithelial-to-mesenchymal (EMT) and mesenchymal-to-epithelial (MET) transformational pathways (Fedele, Cerchia et al. 2019).” – this sentence needs to be re-worked for clarity of its content.

Lines 171-173:” More recent suggested reasons include mesenchymal plasticity exhibited by GB CSCs which are more differentiated and unable to find a suitable niche other than the brain (Ricci-Vitiani, Pallini et al. 2008).” Taking into account the authors refer to a 2008 paper (11 yo), I would suggest changing: “More recent suggested reasons” simply by Another proposed reason …

Lines 237-239: Where it reads: “These findings suggest CSCs within GB and other types of cancers, may be a novel therapeutic target by modulation of the RAS (Roth, Wickremesekera et al. 2019)” it should read: These findings suggest that modulation of the RAS may provide novel therapies targeting CSCs within GB and other types of cancers (Roth, Wickremesekera et al. 2019).

Lines 292-293: “ARBs are associated with statistically improved progression-free survival and overall survival in 81 patients with GB (Januel, Ursu et al. 2015).” – has this study been taken forward with a large cohort of patients or is currently being undertaken? If so, this should be completed with more recent data/publications.

I would like to suggest the authors to include and discuss the following research paper in their section of repurposed drugs for HGG treatment:

Mihajluk K, Simms C, Reay M, Madureira PA, Howarth A, Murray P, Nasser S, Duckworth CA, Pritchard DM, Pilkington GJ, Hill R. “IP1867B suppresses the insulin-like growth factor 1 receptor (IGF1R) ablating epidermal growth factor receptor inhibitor resistance in adult high grade gliomas.” Cancer Lett. 2019 Aug 28;458:29-38. doi: 10.1016/j.canlet.2019.05.028.

Author Response

1) The authors use the old nomenclature, namely glioblastoma instead of the new nomenclature, high grade glioma, this should be changed throughout the manuscript.
We have highlighted the recent revised WHO classification system of astrocytic tumours in the opening paragraph of the introduction section. We believe the reviewer is referring to the old nomenclature ‘glioblastoma multiforme (GBM)’. ‘Glioblastoma’ is the new nomenclature, according to the 2016 World Health Organisation Classification of Tumours of the Central Nervous System (Louis et al) which classifies WHO grade II, III, IV astrocytic tumours as diffuse gliomas, and grade IV astrocytoma as glioblastoma. ‘High grade-glioma’ is not specific as it refers to both grade III and grade IV tumours. Respectfully, we believe that it is appropriate to retain the term “glioblastoma’ which is the focus of our manuscript.
2) Lines 35-37: Where it reads: “Glioblastoma (GB), the most aggressive astrocytic tumor, classified as a grade IV astrocytoma, is characterized by microvascular proliferation and palisading necrosis.” It should read: High grade glioma (HGG), the most aggressive astrocytic tumor, classified as a grade IV astrocytoma, is characterized by microvascular proliferation and a palisading invasive front.
Please see our response to the comment above.
3) Lines 155-157: “It presents an alternative mechanism local recurrence of GB, implicating epithelial-to-mesenchymal (EMT) and mesenchymal-to-epithelial (MET) transformational pathways (Fedele, Cerchia et al. 2019).” – this sentence needs to be re-worked for clarity of its content.
We thank the reviewer for the helpful comment. We have amended the sentence to: ‘It presents an alternative mechanism to explain local recurrence of GB, implicating epithelial-to-mesenchymal (EMT) and mesenchymal-to-epithelial (MET) transformational pathways (Fedele, Cerchia et al. 2019), a paradigm counterintuitive to the concept of activation of regional non-circulating quiescent GB CSCs causing local recurrence of GB.’
4) Lines 171-173:” More recent suggested reasons include mesenchymal plasticity exhibited by GB CSCs which are more differentiated and unable to find a suitable niche other than the brain (Ricci-Vitiani, Pallini et al. 2008).” Taking into account the authors refer to a 2008 paper (11 yo), I would suggest changing: “More recent suggested reasons” simply by Another proposed reason …
We thank the reviewer for the helpful suggestion and we have amended the sentence accordingly.
5) Lines 237-239: Where it reads: “These findings suggest CSCs within GB and other types of cancers, may be a novel therapeutic target by modulation of the RAS (Roth, Wickremesekera et al. 2019)” it should read: These findings suggest that modulation of the RAS may provide novel therapies targeting CSCs within GB and other types of cancers (Roth, Wickremesekera et al. 2019).
We have amended the sentence.
6) Lines 292-293: “ARBs are associated with statistically improved progression-free survival and overall survival in 81 patients with GB (Januel, Ursu et al. 2015).” – has this study been taken forward with a large cohort of patients or is currently being undertaken? If so, this should be completed with more recent data/publications.
This paper suggests that further prospective clinical trials are needed to demonstrate the use of renin-angiotensin system blocker (ARB) to improve clinical outcomes for patients with glioblastoma. Two authors from this paper (R. Ursu, A.F. Carpentier) have published a paper in March 2019 ‘Angiotensin II receptor blockers, steroids and radiotherapy in glioblastoma – a randomised multicentre trial (ASTER trial). An ANOCEF study’ demonstrating that the addition of losartan (an ARB) to the standard of care did not have any impact on steroid requirements or show any statistical differences in reduction in peritumoural oedema on MRI over time. This study has been referred to in the 1st paragraph of Section 5 ‘Repurposing Drugs that Target the RAS’.
7) I would like to suggest the authors to include and discuss the following research paper in their section of repurposed drugs for HGG treatment:
Mihajluk K, Simms C, Reay M, Madureira PA, Howarth A, Murray P, Nasser S, Duckworth CA, Pritchard DM, Pilkington GJ, Hill R. “IP1867B suppresses the insulin-like growth factor 1 receptor (IGF1R) ablating epidermal growth factor receptor inhibitor resistance in adult high grade gliomas.” Cancer Lett. 2019 Aug 28;458:29-38. doi: 10.1016/j.canlet.2019.05.028.
We thank the reviewer for highlighting this. This article has been included in the 3rd paragraph of Section 5 ‘Repurposing Drugs that Target the RAS’. “More recently, a reformulated “liquid” aspirin (IP1867B) has been shown to be a putative IGF and IGFR-1 (signalling pathways in RAS) inhibitor with reduction in high grade glioma tumour burden and improved gastric side-effect profile (Mihajluk, Simms et al. 2019).”.

Reviewer 2 Report

The review comprehensively describes glioblastoma, glioma stem cells, and Angiotensin-Renin system. It is overall well-written as well as the topic is relevant and of interest for a broad readership. Anyway, the current quality of this manuscript is not adequate to warrant publication on Cells. To strengthen the readability of the paper, I believe that the following points should be addressed:

Some links between concepts, paragraphs and sections are missing. The link between GSCs and RAS should be more evident in the review, while it is mentioned just at line 235 and 266. Also the link between EMT and RAS should be better explained. The title of the chapter 3 has to be changed. Circulating cancer stem cells is a minor part of this chapter. In the introduction page 2 the authors mentioned the 4 GBM subtypes while it is now well admitted that there are actually 3 subtypes. The paragraph “Glioma cancer stem cell” is not well written.  The flow of this paragraph is not clear and it seems that the authors are jumping from one idea to another without clear links. (e.g they are focusing on the embryonic stem cell markers but without talking about the other more specific to GBM). Plasticity of GSCs should be mentioned. A table summarizing the publications on targeting the RAS in GBM would help to sup-up the part 5. Logic transition between line 239 and 241, and between line 259 and 266 is lacking. Reference format at line 304. The pink cells are not well understandable in Figure 2. Are they cells? if so, why the transmembrane receptor cross them?

Author Response

1) Some links between concepts, paragraphs and sections are missing. The link between GSCs and RAS should be more evident in the review, while it is mentioned just at line 235 and 266.

The association between GSCs and RAS is summarised in Figure 2, and their relationship has been detailed in the paper, particularly in Section 4 ‘The Renin-angiotensin System’.

2) Also the link between EMT and RAS should be better explained.

This has been expanded in the 5th paragraph of Section 3 ‘Circulating Cancer Stem Cells & Epithelial-to-Mesenchymal Transition’, using colorectal cancer as an example. “The regulatory effect of epithelial-to-mesenchymal transition by the renin angiotensin system (RAS) has been demonstrated in colorectal cancer. In one study, angiotensin II was shown to induce migration of colorectal cancer cells via angiotensin II receptors 1 and 2. Effects mediated by angiotensin II receptor 1 were associated with changes typical of epithelial-to-mesenchymal transition, namely increased expression of E-cadherin, reduced ZEB1 and vimentin levels (Nguyen, Ager et al. 2016).”.

3) The title of the chapter 3 has to be changed. Circulating cancer stem cells is a minor part of this chapter.

We thank the reviewer for the suggestion. The title of this section has been changed to ‘Circulating Cancer Stem Cells and Epithelial-to-Mesenchymal Transition’.

4) In the introduction page 2 the authors mentioned the 4 GBM subtypes while it is now well admitted that there are actually 3 subtypes.

We thank the reviewer for pointing this out. It has now been amended to 3 subtypes. Relevant paper has been cited. “Following the original four GB-subtype classification by the Cancer Genome Atlas Network in 2010 (Verhaak, Hoadley et al. 2010), recent papers have reported refinement of the classification into three GB-subtypes, namely Classical, Mesenchymal and Proneural/Neural (Shen, Mo et al. 2012, Wang, Hu et al. 2018).”.

5) The paragraph “Glioma cancer stem cell” is not well written. The flow of this paragraph is not clear and it seems that the authors are jumping from one idea to another without clear links. (e.g they are focusing on the embryonic stem cell markers but without talking about the other more specific to GBM). Plasticity of GSCs should be mentioned.

Plasticity of GSCs is further discussed in the 1st paragraph of Section 2 ‘Glioblastoma Cancer Stem Cells’.

6) A table summarizing the publications on targeting the RAS in GBM would help to sum-up the part 5.

We thank the reviewer for the suggestion. Table 2 has been added to the end of Section 5 to summarise the current publications.

7) The pink cells are not well understandable in Figure 2. Are they cells? if so, why the transmembrane receptor cross them?

The pink cells are endothelial cells and it is a cartoon meant to show localisation of components of the renin-angiotensin system (i.e. to the endothelial cells of the microvessels as stated in Figure 2). We have highlighted this in the figure legend.